# What Is Hidden in Patients with Unknown Nephropathy? Genetic Screening Could Be the Missing Link in Kidney Transplantation Diagnosis and Management

**DOI:** 10.3390/ijms25031436

**Published:** 2024-01-24

**Authors:** Adele Mitrotti, Ighli Di Bari, Marica Giliberti, Rossana Franzin, Francesca Conserva, Anna Chiusolo, Maddalena Gigante, Matteo Accetturo, Cesira Cafiero, Luisa Ricciato, Emma Diletta Stea, Cinzia Forleo, Anna Gallone, Michele Rossini, Marco Fiorentino, Giuseppe Castellano, Paola Pontrelli, Loreto Gesualdo

**Affiliations:** 1Department of Precision and Regenerative Medicine and Ionian Area (DIMEPRE-J), University of Bari Aldo Moro, 70124 Bari, Italy; adele.mitrotti@uniba.it (A.M.); ighli.dibari@uniba.it (I.D.B.); gilibertimarica@gmail.com (M.G.); rossana.franzin@uniba.it (R.F.); francesca.conserva@uniba.it (F.C.); annachiusolo@hotmail.it (A.C.); mgigante72@gmail.com (M.G.); matteoaccetturo@yahoo.it (M.A.); c.cafiero@yahoo.it (C.C.); luisaricciato1993@gmail.com (L.R.); emmadiletta.stea@gmail.com (E.D.S.); cinzia.forleo@uniba.it (C.F.); michelerossini@libero.it (M.R.); marco.fiorentino@uniba.it (M.F.); 2Department of Basic Medical Sciences, Neurosciences and Sense Organs, University of Bari Aldo Moro, 70121 Bari, Italy; anna.gallone@uniba.it; 3Department of Clinical Sciences and Community Health, University of Milano, 20122 Milano, Italy; giuseppe.castellano@policlinico.mi.it; 4Fondazione IRCCS Cà Grande Ospedale Maggiore Policlinico, 20122 Milano, Italy

**Keywords:** CKD, unknown ESRD, FSGS, genetic testing, next-generation sequencing

## Abstract

Between 15–20% of patients with end stage renal disease (ESRD) do not know the cause of the primary kidney disease and can develop complications after kidney transplantation. We performed a genetic screening in 300 patients with kidney transplantation, or undiagnosed primary renal disease, in order to identify the primary disease cause and discriminate between overlapping phenotypes. We used a custom-made panel for next-generation sequencing (Agilent technology, Santa Clara, CA, USA), including genes associated with Fabry disease, podocytopaties, complement-mediated nephropathies and Alport syndrome-related diseases. We detected candidate diagnostic variants in genes associated with nephrotic syndrome and Focal Segmental Glomerulosclerosis (FSGS) in 29 out of 300 patients, solving about 10% of the probands. We also identified the same genetic cause of the disease (*PAX2*: c.1266dupC) in three family members with different clinical diagnoses. Interestingly we also found one female patient carrying a novel missense variant, c.1259C>A (p.Thr420Lys), in the *GLA* gene not previously associated with Fabry disease, which is in silico defined as a likely pathogenic and destabilizing, and associated with a mild alteration in GLA enzymatic activity. The identification of the specific genetic background may provide an opportunity to evaluate the risk of recurrence of the primary disease, especially among patient candidates living with a donor kidney transplant.

## 1. Introduction

Chronic kidney disease (CKD) is now considered to be a global public health issue due to its rising prevalence, between 7–12% worldwide [1], and its progression to end-stage renal disease (ESRD). In the last decade, many studies identified several genetic and molecular processes associated with kidney failure. Quaglia M. et al. described an unexpectedly high prevalence of rare genetic disorders in patients with ESRD of unknown origin [2], suggesting that further data need to be analyzed, with the ultimate goal of attaining more specific precision-medicine approaches [3]. About 20% of Europeans affected by ESRD have a missing/unknown diagnosis [4]. Monogenic kidney diseases account for a large proportion of CKD patients. It has been estimated that while, as expected, 70% of pediatric patients with ESRD have a hereditary cause, 10 to 15% of CKD phenotypes in the adult population can be explained by genetic mutations [5,6,7,8]. Those data suggest that monogenic disease associated to kidney disorders in adults, remain often underestimated.

As demonstrated by a large cohort of patients affected by CKD, genetic testing may allow a reclassification of the initial kidney disorder [8,9,10]. When a hereditary disorder is identified, it may help in the selection of treatment options, as genetically caused nephropathy does not respond to immunosuppressive therapy. Furthermore, patients with positive family history for genetic disorders may benefit from kidney transplant planning, especially when living donors are available.

As recently reported by the ERA Working Group on Inherited Kidney Disorders (WGIKD), it has been that suggested that CKD/ESRD patients with an unclear genetic diagnosis, and especially those younger than 50 years-of-age, should perform a first genetic approach, and establish a large multigene targeting panel that includes potential causative genes. In this case, exome/Genome sequencing would be reserved only for unsolved cases with a strong family history or medical records that suggest an underlying genetic condition [11]. Thus, even if the advent of high throughput sequencing technology would privilege the analysis of the entire coding DNA material through exome or genome sequencing, both the higher complexity of data analysis and the need for more expensive sequencers and specialised personnel may limit these technologies to specific centers. Next-generation sequencing (NGS) is a technology for DNA and RNA sequencing and variant detection that can sequence hundreds or thousands of genes or entire genomes in a short time period. Sequence variants detected by NGS are widely used for disease diagnosis, prognosis, therapeutic decision-making, and patient follow-up. NGS has the ability to produce a huge amount of data and offers a highly efficient, low-cost, rapid and accurate method of DNA sequencing, suggesting its possible use as a primary sequencing approach.

Genetic testing can also be useful in the case of phenocopies or diseases with different phenotypes as a result of different pathogenic drivers. For example, while focal segmental glomerulosclerosis (FSGS) does not represent a single disease entity, it may include a wide spectrum of conditions with different clinical-pathological patterns and important clinical implications, with important implications for treatment choice and prognostic evaluations [12]. The genetic study of patients with FSGS is therefore useful for a more precise etiopathogenetic definition, and in particular for patients who are candidates to receive a kidney transplant, especially from a living donor, bearing in mind the risk of recurrence of the disease after transplantation. Moreover, patients with ESRD on dialysis and/or patients subjected to kidney transplant may show a hidden reduction of plasma α-galactosidase A (GLA) activity [13] that masks Fabry disease (FD), and this also applies to cases where kidney biopsies may manifest features of focal segmental glomerulosclerosis (FSGS), which leads to Fabry disease being misdiagnosed in these cases [14].

In the present study, we aimed to explore if genetic screening by NGS could help to identify the primary kidney defect and solve some misdiagnosed genetic disorders associated with CKD/ESRD and FSGS, with the aim of supporting clinical decisions, especially for patient candidates with a living donor kidney transplant. Although large-scale sequencing technologies have, over the last decade, become the largest methods used to assess DNA-sequencing analysis, we also demonstrated that NGS custom-panel, if properly designed, may provide a powerful tool and proper modality that can be used to identify otherwise unknown genetic causes of kidney diseases. When whole or exome sequencing are not available, NGS panel can still be considered to be consistent with the final aim of addressing the molecular hidden genetic defect.

## 2. Results

### 2.1. Pathogenic Variants in Genes Related to Nephrotic Syndromes (NS) Solved 10% of the Cohort

The study population was selected from a group of 4087 patients who attended the Nephrology outpatient clinic who were affected by CKD or ESRD, who did not have a histological diagnosis of causal nephropathy, and a biopsy-proven idiopathic FSGS. On the basis of these clinical criteria, we screened 787 patients; of these, 387 patients were on a waiting list for living or cadaveric kidney transplantation, 150 had already undergone kidney transplant, 50 patients were recruited from a CKD outpatient clinic, and 200 were recruited from the Italian Renal Registry of Kidney Biopsies (IRRB). Ultimately, a total of 300 patients were enrolled (Figure 1). This selected cohort of patients included 118 females (39%) and 182 males (61%), of whom 298 were white (not Hispanic or Latin ethnicity), and the other two of Hispanic or Latin ethnicity; a total of 162 patients (54%) had a biopsy-proven diagnosis of idiopathic FSGS, and a smaller number (138) had CKD/ESRD of unknown origin (46%). The mean age of kidney disease onset was 35.65 ± 16 for all patients included in the study; this increased to 37.17 ± 15.5 for patients with unknown CKD/ESRD; and fell to 32.23 ± 16.9 for patients with a biopsy-proven diagnosis of idiopathic FSGS. A total of 130 patients h ad already received a kidney transplant before the enrollment, and 23 patients reported a positive family history of nephropathy (Table 1).

The mean age of kidney disease onset was: 35.65 ± 16 for all patients included in the study; this was higher (37.17 ± 15.5) for patients with unknown CKD/ESRD and lower (32.23 ± 16.9) for patients with a biopsy-proven diagnosis of idiopathic FSGS. All patients underwent genetic analysis in NGS. Our NGS custom panel included 63 genes involved in structural or functional molecular pathways that potentially lead to CKD, ESRD and FSGS (Appendix A). We detected candidate diagnostic variants in 29 of the 300 (about 10%) probands. Table 2 lists all of the candidate diagnostic variants that were, on the basis of ACMG criteria classified, as (a) Pathogenic (P), (b) Likely Pathogenic (LP) or, when a complete pathogenicity meaning needed more investigation, as (c) Variant of Unknown Significance (VUS)/Likely Pathogenic (LP). In our cohort we identified nine patients (3%) (ID:46, 57, 63, 67, 77, 89, 176, 232, 237) with variants in Collagen Type IV αgenes (*COL4A*), including *COL4A3*, *COL4A4*, *COL4A5*. Half of the missense found in *COLA4* genes affect a glycine residue. In a single patient (ID:176), we found a compound (heterozygous pathogenic mutations in *COL4A4*), while another (ID:67) showed a pathogenic mutation in *COL4A4* (2-227922281-C-A), as well as a heterozygous VUS in *MYOE1* (15-59430471-A-G) that is located in a strongly conserved position (phyloP100way = 9.32 is greater than 7.2), absent in gnomAD and is also, on the basis of a 10 score of deleteriousness, predicted to be pathogenic (BayesDel_addAF, DANN, EIGEN, FATHMM-MKL, LIST-S2, M-CAP, MutationAssessor, MutationTaster, PrimateAI and SIFT); however, further investigations are needed to express its significance.

We identified homozygous pathogenic and very rare variants in structural/functional genes responsible for nephrotic syndrome in another 14 patients (ID: 51, 62, 77, 164, 168, 169, 182, 196, 198, 219, 232, 235, 243, 244), eight of whom reported as unknown CKD or unknown ESRD: a total of seven presented truncating mutations (ID: 62, 77, 196, 232, 235, 243, 244) and the others showed missense variants. We therefore identified a diagnostic genetic cause in 19 patients who had an otherwise unknown kidney disease (Table 2).

In an additional five patients (ID:48, 60, 92, 170, 191), we identified interesting variants of nephrotic syndrome recessive genes with potential diagnostic meaning (Table 3), but which will requires further investigations, whether through a segregation test of affected family members or functional studies. NGS analysis revealed an homozygous frameshift variant in the *PAX2* gene [MIM: 616002], c.1266dupC, (p.Asp423ArgfsTer84) defined as LP according to ACMC criteria, in one patient affected by idiopathic FSGS (ID:223). On the basis of the positive family history record, we investigated if the same variant was shared by other affected family members, and we were interested to find that the same *PAX2* pathogenic mutations were shared by affected family members, even if heterogeneous phenotypes clinically diagnosed as FSGS, Membranoproliferative glomerulonephritis type II (MPGN II) and unknown ESRD (Table 4) presented themselves. Our index (ID:223) is a 40 year-old man who presented enuresis until 17 years-of-age. Renal biopsy was performed when he was 27 years-old, in response to new onset proteinuria and worsening renal function. The histological diagnosis was FSGS (NOS variant). His sister (patient ID:224) is a 52-year-old woman. When she was 27 years-old, she had a renal biopsy diagnosis of MPGN II. After 6 years, she started haemodialysis and, at the age of 36 years-of-age, received a kidney from a deceased donor. His brother (patient ID:225) is a 49 year-old man, who started haemodialysis at the age of 22 years without a histological diagnosis of kidney disease before, after six years, receiving a kidney from a deceased donor. Thus, different phenotypes and clinical evaluations subtended the same genetic variation.

### 2.2. Genetic Analysis of the GLA Variants

Renal pathology features of FSGS can also be identified in patients with Fabry disease, and so we focused our attention on the analysis of all variants located in the coding regions of the *GLA* gene (NM_000169.3). Gene sequencing analysis identified two exonic missense variants: c.937G>T, (p.Asp313Tyr), which was found in two different patients (ID:5, 211); and the variant c.1259C>A (p.Thr420Lys), which was found in patient ID:167. We identified two additional variants of unknown significance (VUS), which were located into the 5′ upstream regulatory region of the gene: c.-10C>T was found in 14 different patients (ID:7, 11, 38, 41, 43, 46, 133, 148, 152, 169, 176, 190, 192, 211); and c.-12G>A was found in 7 different patients (ID:65, 93, 153, 217, 223, 224, 226). All described variants of the *GLA* gene are listed in Table 5.

The exonic variant c.1259C>A,(p.Thr420Lys), which was found in one female patient, is an heterozygous missense mutation located on exon 7 of the *GLA* gene 16-100652828-G-T. According to the ACMG criteria [15], this variant is classified as VUS/LP (ACMG criteria: PM2, PM1, PP2 and BP4). This variant is located very close to the α galactosidase A C-terminal beta sandwich domain, and falls into an exonic hotspot on exon 7 (as listed on the Fabry-Database.org—see http://fabry-database.org/ (accessed on 21 December 2023). Thus, even when the ACMG criteria is referred to, it is reported as VUS, and the presence of multiple scores of pathogenicity (PM1, PP2, PM2, BP4), suggested a potential role in altering the function of the encoding protein, with possible or even likely Pathogenic meaning. For this reason, we considered the variant VUS/LP to be of particular interest. This is a novel variant, which is not reported in the online-available genome and exome sequencing database as gnomAD, and is not listed along the 900 disease-causing *GLA* variants grouped in the freely accessible Fabrydatabase (http://fabry-database.org/ (accessed on 21 December 2023). The MetaLR score [16,17], which is based on a bioinformatic tool that uses logistic regression to combine eleven independent pathogenicity scores (including the Genomic Evolutionary Rate Profiling (GERP)) [18], is 0.96, and associated with a damaging meaning; to our knowledge, however, there are no published functional studies that demonstrater the effect of this rare missense variant on protein structure and function. I In order to evaluate the impact of this mutation on protein folding and, as a consequence, its activity, we used a computational approach developed by DynaMut [19], and found that the *GLA*: p.Thr420Lys variant induces a difference between interactions, along with surrounding residues involved on any type of interactions (Figure 2A,B). Moreover, after assessing the impact of this mutation on protein dynamics and stability, we obtained a prediction outcome score ΔΔG = −0.831 kcal/mol (which the software indicated to be destabilizing) (Figure 2C). The same results were also confirmed by two other different tools (mCSM [20] and DUET) [21], which again predicted a destabilizing effect of the identified variant (ΔΔG mCSM: −0.870 kcal/mol-Destabilizing; ΔΔG DUET: −0.474 kcal/mol-Destabilizing).

The patient carrying this variant is a 70year-old Caucasian female with a negative family history of kidney and heart diseases. Clinical features included hypertension and progressive CKD leading to ESRD. She started haemodialysis at the age of 38 years-of-age and obtained a kidney from a deceased donor at the age of 54 years-of-age, before developing delayed graft function. The patient also had osteoporosis and heart valve disease, presenting moderate mitral valve stenosis, mild aortic valve stenosis, along with ectasia of ascending aorta and left ventricular (LV) hypertrophy, with an interventricular septum of 12 mm and an ejection fraction (EF) of 60%.

The measurement of α-Gal enzyme activity revealed a value of 8.7 nmol/mL/h. Random skewing of X-inactivation in females can explain a possible mild reduction of encoded protein expression levels, varying from 25 to 75 percent of normal enzyme activity; more severe or non-random skewing, in contrast, can cause expression levels to vary less than 25 percent [22]. Compared to the wild type patient (>15 nmol/mL/h), the value of 8.7 nmol/mL/h, is about 50% reduced, suggesting that this variant could play a role in the enzymatic activity of GLA protein. We also performed lyso-Gb3 concentration analysis on this patient: a reduced value of 1.53 nmol/L was recorded, compared against the normal value of lyso-Gb3 are ≤2.3 nmol/L. Smid et al. suggested that, in uncertain cases, increased lyso-Gb3 values are very suggestive of Fabry disease, but added that normal values cannot exclude this [23]. Familial co-segregation studies were not possible because family members were unwilling to participate in the study.

In other two unrelated patients, we identified another missense variant of the *GLA* gene on exon 6, X-100653420-C-A, c.937G>T, (p.Asp313Tyr), which has already been described in the literature [24]. This variant was also responsible for altered interactions between surrounding residues (Figure 2D,E). We assessed the impact of this mutation on protein dynamics and stability, and obtained a prediction outcome score of ΔΔG: 0.617 kcal/mol (indicated by the software as stabilizing), along with a decrease of molecule flexibility on the blue side of Dynamut (Figure 2F), which was confirmed by mCSM (ΔΔG mCSM: 0.177 kcal/mol-Stabilizing) and DUET (0.033 kcal/mol-Stabilizing) [20,21].

According to the ACMG criteria, this variant is classified as LP [ACMG criteria: PM1, PP2, PP3, PP5] [25]. The first patient carrying the variant c.937G>T, (p.Asp313Tyr) is a 65 year-old female affected by hypertension who had recently died after heart failure. Her medical records reported progressive CKD of unknown origin and hypertension. She denied she had a positive family history of nephropathy and cardiovascular or cerebrovascular events. She started hemodialysis at 56 years-of-age and then underwent deceased donor kidney transplantation.

The second patient who carried the *GLA* gene c.937G>T, (p.Asp313Tyr) variant was a 56 year-old female with nephrotic syndrome who underwent a kidney biopsy when she was 45 years-old. Renal pathology showed an FSGS pattern of injury. Past medical records reported hypertension and an increased level of homocysteine, as well as an early diagnosis of mitral prolapse. Her father died from cerebrovascular disease. She reported her uncle on her mother’s side started hemodialysis at 60 years-of-age, and her aunt on her father’s side underwent deceased donor kidney transplantation. Both family members had ESRD of unknown origin.

In addition we found a *GLA* variant X-101407913–G-A, (c.-10C>T), located in the very early sequence of chromosome X, in 14 unrelated indexes. This variant is placed in the 5′ UTR region of the GLA gene and is, according to the ACMG criteria, classified as benign. All these patients presented a past medical history of CKD; three patients showed a histological diagnosis of idiopathic FSGS; two had developed arrhythmia; one had a diagnosis of myocardial hypertrophy; and the remaining eleven patients, ten of whom had a kidney transplant, had CKD/ESRD of unknown origin, All patients with this variant showed no clinical signs nor symptoms of central and peripheral nervous system involvement.

In another seven unrelated patients, we discovered another variant of chromosome X-100662903-G-A, (c.-12G>A), which located in the 5′UTR region of the *GLA* gene. According to the ACMG criteria, it is however classified as benign. All seven patients had kidney disease and three had histological diagnosis of FSGS and a family history of nephropathy (one with valve abnormalities). one patient had a family history (two brothers with ESRD), one was on dialysis and another patient had a kidney transplant.

In acknowledging the heterogenity and complex genetics of FD, we decided to investigate if any *GLA* variants could possibly detect any α-Gal enzyme reduction. A total of 14 (out of 23 patients) carrying exonic variants of the GLA gene were subjected to DBS to test the enzymatic activity of GLA. Three male patients carrying the variant c.-10C>T, along with other variants defined as benign by ACMG criteria, reported a slight decrease in enzymatic activity (ID7: 13.3 nmol/mL/h; ID38: 14.5 nmol/mL/h; ID133: 11.5 nmol/mL/h). One male patient (ID169) had a α-Gal enzyme activity of 7.2 nmol/mL/h.

## 3. Discussion

A total of 17% of patients with ESRD around the world do not have a primary renal disease diagnosis and are thus classified as CKD of undetermined aetiology [26]. The absence of a diagnosis, as well as a misdiagnosis, can have therapeutic consequences [27]. In recent years, there has been a lot of progress in the genetic testing of kidney disease, which has helped us to learn more about the genes involved in kidney disease. The majority of inherited kidney diseases are linked to a wide range of phenotypes and exhibit high levels of genetic heterogeneity, which can make it hard to classify kidney diseases based on phenotype, and lead to a wrong diagnosis [27]. NGS-based approaches could enhance diagnostic accuracy in individuals with CKD of unknown cause. Moreover, the use of genetic testing can be particularly useful in the selection of kidney donors, as well as in transplant recipient management [28], since it can support the clinician when they are evaluating recurrence risk and selection of living donors.

In this manuscript we described the experience of our center in identifying the genetic cause of the disease in an Italian cohort of 300 patients from the south of Italy with CKD/ESRD of unknown aetiology andbiopsy-proven evidence of idiopathic FSGS.

In our cohort we identified *COL4A* variants including *COL4A3*, *COL4A4*, *COL4A5*, in nine patients, which mainly affect glycine residue in patients with a diagnosis of idiopathic FSGS. Glycine missense mutations in type IV collagen genes have been reported to deeply affect triple helix formation and eventually correlate with phenotype severity [29]. *COL4A3-5* genes, which are classically associated with Alport syndrome, are now understood to also be involved in the aetiology of focal segmental glomerulosclerosis, and NGS approaches thus enabled an expansion and redefinition of genetic kidney disease categories, suggesting that the diagnoses should be made on the basis of clinical evaluation as well as genetic data [30]. NGS-based techniques improved diagnostic accuracy in patients with CKD of unknown origin, with a yield ranging from 12 to 56% throughout the different cohorts [11]. These techniques also improved patient care, since NGS has the potential to identify the cause of CKD at an early stage of the disease, which allows for timely intervention to delay or prevent ESRD [27]. Interestingly in our patient cohort we identified pathogenic variants in 11 (3.7%) patients, LP were found in 19 (6.3%) patients, and a total of 29 patients (10%) received a definitive diagnosis based on genetic approaches. Further analysis will be performed on variant segregation in families in order to definitively confirm their pathogenic role.

Moreover, our data also demonstrated that NGS panels are also useful to enable a more precise differentiation of phenocopies and reclassification of the primary diagnosis in individual patients. One example was provided by the described three members of the same family with the homozygous frameshift variant in *PAX2*: c.1266dupC, who showed three different clinical phenotypes (Table 4). These results indicated that specifically designed kidney disease gene panels are very useful, especially in the case of diseases that can manifest with different phenotypes or diseases that are caused by -variants in different genes, such as cystic inherited or glomerular kidney diseases. In these cases, the use of specifically designed kidney disease gene panels can identify the genetic cause of the disease in up to 78% of patients with a suspected clinical disease [31].

It has been described that a phenocopy of FSGS can be due to mutation in the GLA gene [32], and we therefore also focused on GLA variants when engaging our cohort. We identified an interesting novel missense mutation in the *GLA* gene c.1259C>A, (p.Thr420Lys), which was absentboth from the scientific literature and gnomAD in a patient with a clinical diagnosis of unknown ESRD. In the Franklin by Genoox database, this variant has been recently associated with angiokeratoma corporis diffusum, a clinical symptom of Fabry disease. Bioinformatical tools and computational models confirmed that this variant could affect protein folding and cause reduced activity of the GLA enzyme. These in silico algorithms have been recommended by the American College of Medical Genetics and American College of Pathologists as variant classification guidelines for clinical reporting in diagnostic laboratories for variant interpretation [15,33]. Moreover, in our patient carrying this variant we also observed a 50% reduction in enzymatic activity, when compared to control patients. Clinically this female patient showed some clinical features that are known to be associated with Fabry disease, such as bone involvement [34]. Indeed, secondary osteoporosis can be associated with decreased intestinal vitamin D absorption, which is in turn associated with several mechanisms that precede the onset of Fabry disease. Moreover, this patient shows cardiology issues and valvular heart disease, which are also present in Fabry disease patients. 

The variant c.1259C>A (p.Thr420Lys) involves a Threonine residue that is a relatively high-evolving residue [35]. By using specific algorithms, the prediction we made of protein stability and dynamics showed a clear destabilization of the protein. In order to specifically assess the pathogenic role of this variant, further functional studies are needed, including, for example, in vitro mutagenesis and functional evaluation. Moreover, it would be very interesting to perform WES studies on our patient, in order to exclude other genetic factors that can be responsible got the described phenotype.

We also identified the variant c.937G>T, (p.Asp313Tyr), alias (D313Y), which has been extensively debated because of conflicting interpretations of pathogenicity and phenotypic manifestations. The D313Y variant cannot really represent the causative cause of Fabry disease in all the cases [24], although it has been described as the most frequent genetic alterations reported in European asymptomatic newborns [36] and, in some cases, as being associated with Fabry symptoms [37]. In our cohort we further detected several intronic variants in the *GLA* gene. The importance of intronic variants in the milieu of Fabry disease is emerging [38]. We identified two intronic variants in the *GLA*, at position c.-10C>T and c.-12G>A, in 12 and 5 patients respectively. Both are located in the promoter region of the *GLA* gene and could be implicated in decreasing GLA expression at the transcription and/or translation levels [39,40,41]. By analysing intronic variants, we also identified (in one patient) the presence of four previously reported variants that were combined in a heterozygote haplotype (-10C>T, c.370-77_-81del, c.640-16A>G, c.1000-22C>T) involved in mapping promoter and regulatory intronic regions. These intronic mutations could play both qualitative and quantitative roles in the transcription of the gene and in the translation of α-galactosidase A. This hypothesis is supported in some cases by the presence of Gb3 and/or lyso-Gb3 in the blood and urine of patients [41,42]. Moreover, panel-based approaches do not allow for the complete evaluation of deep intronic variants that could also play a role in the modulation of gene expression.

In conclusion, we settled up a genetic panel based on NGS that is able to identify DNA mutations in patients with ESRD of unknown origin, and also inform patients with inconclusive kidney biopsy about the pathogenesis of the disease. In the context of genomic medicine, NGS techniques have the potential to improve the diagnostic efficiency of genetic renal diseases, re-classification of kidney diseases, diagnosis of early-onset CKD, and also (in adults) prevent the progression of end-stage renal disease [8,30]. Moreover, the identification of the specific genetic background may provide an opportunity to evaluate the risk of the primary disease recurring, especially in patient candidates with a living donor kidney transplant.

The abundance of genetic and molecular information generated by next-generation sequencing poses a new challenge, due to the growing needs of efficient model systems and of bioinformatic capacities that could translate to improved precision diagnostics and aid the prognosis and long-term management of kidney disease. In achieving this goal, the custom panel can, compared to much more expensive sequencing methods, more effectively help us focus on and interpret the detection of kidney disease. Ad hoc panels targeted on a particular gene set make it possible to test and analyse known disease-causing genes that can be clinically characterized by different phenotypes. However, the identification of the specific disease-causing gene could be very important, both for prognosis and patient treatment. The midsize-custom designed panel is as efficient as Whole Exome Sequencing (WES) and Whole Genome Sequencing (WGS) in mapping variants of biological and clinical relevance, and renders higher coverage at a lower cost [43].

## 4. Materials and Methods

### 4.1. Patient Population

This is a single center, observational, prospective study. The study population was selected from 4087 patients seen at the outpatient clinic of the Nephrology, Dialysis and Transplantation Unit of the Policlinic of Bari in the period from July 2017 to July 2019. The clinical and research activities are consistent with the Principles of the Declaration of Istanbul. The study was approved by the local ethical committee (Prot. 95286, 14 December 2016, study No. 5003) and was conducted in accordance with the Declaration of Helsinki. Informed consent was obtained from all study subjects y. All patients, after giving informed consent, also provided blood samples for DNA extraction and the collation of demographic and clinical data. Various comorbidities were identified in patients, including diabetes (5%), hypertension (49%), cardio-vascular disease (including myocardial hypertrophy, ischemic heart disease, cardiac valvulopathy, heart failure, peripheral vascular disease and arrhythmias) (18%) and CAKUT anomalies (including congenital solitary kidney, renal cysts, double renal district, vesicoureteral reflux) (4%).

### 4.2. Gene Selection for the Custom-Panel Design

The selection of genes to be included in the panel was made by searching Orphanet (https://www.orpha.net (accessed on 21 December 2023) and OMIM (https://www.omim.org (accessed on 21 December 2023), and all the genes identified as potentially mutated were identified in focal segmental glomerulosclerosis, genetic and hereditary nephrotic syndrome, Alport Syndrome, podocytopathies and Fabry disease. The complete list of selected genes is provided in Appendix A.

### 4.3. DNA Extraction and Next Generation Sequencing (NGS)

DNA was extracted from whole blood samples in EDTA by using the QIAamp DNA mini kit (Qiagen, Hilden, Germany). An NGS custom panel (Agilent Technologies, Santa Clara, CA, USA) was designed, and we used the SureSelectQXT protocol for Illumina Multiplexed Sequencing on MiSeq Desktop Sequencer (Illumina Inc., San Diego, CA, USA). The custom panel covers all exons and flanking intronic regions (+/− 10bp) of 63 genes involved in different renal diseases (Appendix A); however the Agilent SureSelect kits used in this project contain an in-solution capture method that utilizes long 120 mer, biotinylated cRNA baits to enrich regions from genomic DNA fragments. In addition to covering the target regions, this approach also allows off-target binding. The coverage we obtained in each run was more than 90%. The specific uncovered regions were sequenced by using Sanger methods, and the read depth was between 100–200×. The candidates’ pathogenic, or likely pathogenic variants, were also verified by Sanger sequencing (3500DX Genetic Analyzer, Life Technologies, Monza, Italy) of target regions after PCR amplification.

### 4.4. Annotation and Variants Analysis

Fastq.gz output file was analyzed with Sure Call 3.1 (Agilent Technologies) and Variant Calling Files (VCF) were generated, and Reads were aligned to the Genome Reference Consortium build 37 (Human Genome 19). For each candidate variant, we estimated the allelic frequency through public databases as “The Genome Aggregation Database” (gnomAD) (http://gnomad.broadinstitute.org/ (accessed on 21 December 2023) and “The Single Nucleotide Polymorphism Database” (dbSNP, https://www.ncbi.nlm.nih.gov/projects/SNP/ (accessed on 21 December 2023). Multiple tools for prioritization were used to identify and classify the predicted effect of each single nucleotide variant (SNV) or premature termination in the protein outcome. We used American College Medical Genetics (ACMG) criteria to assess if a single nucleotide variant could be considered to be benign (B) or likely benign (LB; a pathogenic (P) or likely pathogenic (LP); or a variant of unknown significance (VUS), [25]. We consulted further freely-accessible resources for variant interpretation, as advocated by Franklin Genoox (https://franklin.genoox.com (accessed on 21 December 2023), the Online Mendelian Inheritance in Man (OMIM) (https://www.omim.org/ (accessed on 21 December 2023) and ClinVar (https://www.ncbi.nlm.nih.gov/clinvar/ (accessed on 21 December 2023). We considered computational verdict of deleteriousness for each candidate variant, referring to pathogenicity and conservation scores, and European (non-finnish) allele frequency. Multiple software tools have been used for annotation, in order to: (a) predict the effect on protein function (ANNOVAR and SnpEff); (b) estimate the minor allele frequency (MAF) in public databases (gnomAD, dbSNP, 1000 Genomes Project); and (c) predict in-silico deleteriousness (dbNSFP) [44]. In order to support the evidence and our interpretation of human variations and phenotype, we consulted public archives of clinical reports, such as ClinVar and the Human Gene Mutation Database (HGMD), and then manually revised the supporting literature. In undertaking sequence interpretation, and seeking to evaluate the functional impact of heterozygous or homozygous variants that diverge from the reference sequence, when identified as single nucleotide variants (snv) or premature termination, splice-site and insertion-deletion mutations, we used software algorithms such as PolyPhen2 [45] and Combined Annotation Dependent Depletion (CADD) [46].

### 4.5. Further Analysis in Patients with Suspected GLA Variants

In 13 out of 23 patients who showed exonic variants in the GLA gene, we measured α-Gal enzyme activity to evaluate a possible hidden diagnosis of Fabry disease. One patient died before performing the test, and nine patients did not provide informed consent. Dried blood spots (DBS) testing was performed after proper informed consent was obtained, and the results were analyzed at the Institute of Biomedical Research and Innovationin Palermo, Italy.

Lyso-Gb3 concentration was tested in a single female patient through mass spectrometry at the Center for Research and Diagnosis of Lysosomal Storage Diseases in Palermo, Italy. Using different computational approaches, we also performed dynamic analysis to explore if the identified GLA variants could produce any protein-folding defects that would eventually increase the interest in the variant. The analysis of the impact of the identified variants on protein folding was obtained through DynaMut (http://biosig.unimelb.edu.au/dynamut/prediction (accessed on 21 December 2023), a specific web server that adds a dynamic element to mutation analysis. We also used the same tool to analyse protein dynamics and stability arising from vibrational entropy changes, and did this by combining graph-based signatures with normal mode dynamics to provide a consensus prediction score.

### 4.6. Statistical Analysis

Data are presented as mean ± standard deviation (SD). The onset age of kidney disease and standard deviations were calculated by applying the Kolmogorov-Smirnov test of normality. Statistical analysis was performed by using the Statistical Package for Social Sciences version 23.0 (SPSS, Inc., Chicago, IL, USA).

## Figures and Tables

**Figure 1 ijms-25-01436-f001:**
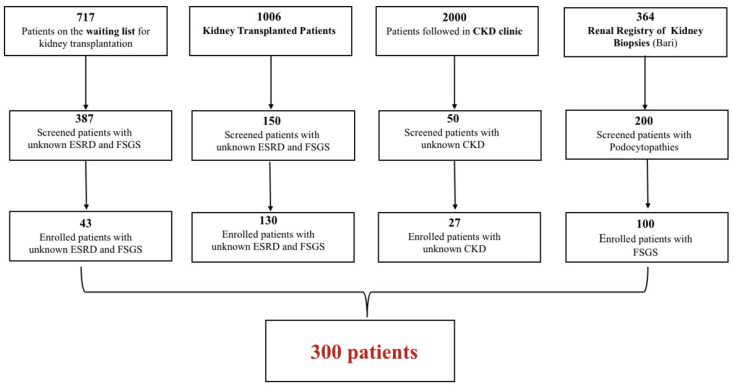
Study workflow.

**Figure 2 ijms-25-01436-f002:**
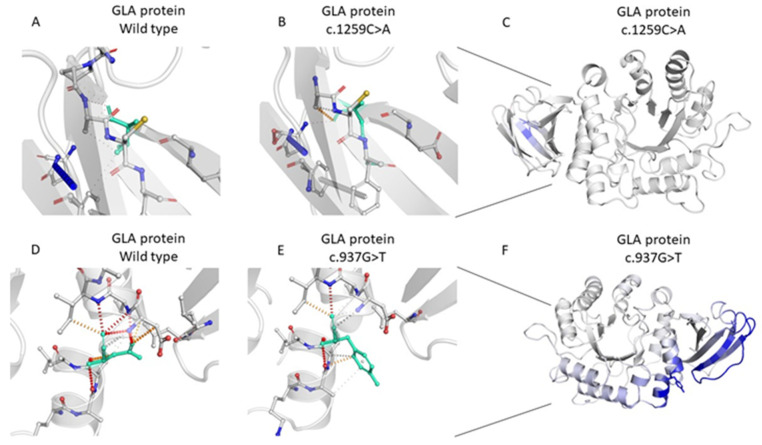
(**A**–**C**). Effect of the variant c.1259C>A, p.Thr420Lys on protein dynamics and stability by DynaMut. Wild-type (**A**) and mutant residues (**B**) are colored in light-green, showing the difference between interactions (along with surrounding residues) involved in any type of interactions. (**C**) Effect of vibrational entropy changes induced by aa variation due to c.1259C>A variant. The blue area represents a rigidification of the structure. (**D**–**F**) The effect of the variant c.937G>T, p.Asp313Tyr (D313Y) on protein dynamics and stability by DynaMut, Wild-type (**D**) and mutant residues (**E**) are colored in light-green, showing the difference between interactions and surrounding residues) involved in any type of interactions. (**F**) Effect of vibrational entropy changes induced by aa variation, due to the c.937G>T variant.

**Table 1 ijms-25-01436-t001:** Clinical and demographic characteristics of the study cohort.

	N. 300
Sex	118 F 182 M
Phenotype	162 Podocytopathies 138 Unknown CKD/ESRD *
Age of Onset of Kidney Disease	35.65 ± 16 Whole cohort 37.17 ± 15.5 CKD/ESRD * 32.23 ± 16.9 FSGS *
Transplant	130 Yes 170 No
Ethnicity	298 Not Hispanic or Latino 2 Hispanic or Latino
Race	298 White 2 Hispanic
Family History	23 Yes 277 No/unknown

* CKD: Chronic kidney disease, ESRD: End Stage Renal Disease FSGS: Focal segmental glomerulosclerosis.

**Table 2 ijms-25-01436-t002:** Diagnostic Pathogenic/Likely Pathogenic variants in structural/functional genes responsible for nephrotic syndrome (NS) that account for 10% of missed diagnoses in our cohort.

Sample	Candidate Gene Name	Chromosome	Genomic Variation	cDNA Variation	dbSNP	Allele Frequency	Genotype	Inheritance	Functional Effect	ACMG Interpretation	Phenotype *
35	*FN1*	2	g.216241356C>T	c.5752G>A	-	0.0	Het	AD	Missense	VUS	Unknown ESRD
46	*COL4A3*	2	g.228173662T>C	c.4510T>C	rs201671013	0.0002350	Het	AD/AR	Missense	VUS/Likely Pathogenic	Unknown ESRD
51	*TTC21B*	2	g.166786784C>T	c.985G>A	rs768266139	0.000008814	Hom	AR	Missense	VUS/Likely Pathogenic	Unknown ESRD
56	*INF2*	14	g.105167927_105167928insAGC	c.226_227insAGC	-	0.0	Het	AD	In-frame insertion	Pathogenic	Unknown ESRD
57	*COL4A4*	2	g.227973547A>G	c.693+2T>C	-	0.0	Het	AD	Splice Donor Loss	Pathogenic	Unknown ESRD
62	*NPHS2*	1	g.179521754_179521755del	c.855_856del	rs749740335	0.0001558	Hom	AR	Frameshift	Pathogenic	Unknown ESRD
63	*COL4A5*	X	g.107827760_107827761insTT	c.1032+16_1032+17dup	-	0.0	Het	AD/AR	Splicing Variant	VUS/Likely Pathogenic	Unknown ESRD
67	*COL4A4*	2	g.227922281C>A	c.2419G>T	-	0.0	Het	AD/AR	Missense	VUS/Likely Pathogenic	FSGS
	*MYO1E*	15	g.59430471A>G	c.4316-2A>G	-	0.0	Het	AR	Missense	VUS	
77	*COL4A5*	X	g.107930710A>G	c.4316-2A>G	-	0.0	Hom	AD/AR	Splicing Loss	Pathogenic	FSGS
89	*COL4A5*	X	g.107911582C>G	c.3638C>G	-	0.0	Het	AD/AR	Missense	Pathogenic	Unknown ESRD
103	*PODXL*	7	g.131241029_131241030insGGCGGC	c.89_90insGCCGCC	-	0.0	Het	AD/AR	In-frame insertion	VUS/Likely Pathogenic	Unknown ESRD
164	*LAMB2*	3	g.49162783G>C	c.2623C>G	-	0.0	Hom	AR	Missense	VUS/Likely Pathogenic	FSGS
168	*LMNA*	1	g.156107470G>A	c.1634G>A	rs142191737	0.0005141	Hom	AR	Missense	Likely Pathogenic	Unknown ESRD
169	*NPHS2*	1	g.179526214C>T	c.686G>A	rs61747728	0.03601	Hom	AR	Missense	VUS/Likely Pathogenic/Pathogenic	Unknown ESRD
176	*COL4A4*	2	g.227967540C>T	c.895G>A	rs757578262	0.0	Het	AD/AR	Missense	Likely Pathogenic	Unknown ESRD
		2	g.227985764C>T	c.293G>A	-	0.0	Het	AD/AR	Missense	Likely Pathogenic	
182	*NPHS1*	19	g.36336621G>T	c.1707C>A	-	0.0	Hom	AR	Missense	Pathogenic	Unknown ESRD
190	*INF2*	14	g.105169734G>A	c.610G>A	rs1049200069	0.0	Het	AD	Missense	VUS	FSGS
195	*TRPC6*	11	g.101323799G>A	c.2683C>T	rs121434394	0.0	Het	AD	Missense	Pathogenic	FSGS
196	*EMP2*	16	g.10641396C>G	c.78+1G>C	rs747072310	0.00003103	Hom	AR	Splicing Variant	Likely Pathogenic	FSGS
198	*PLCE1*	10	g.96013971G>A	c.3304G>A	rs763011760	0.000008838	Hom	AR	Missense	VUS/Likely Pathogenic	FSGS
212	*WT1*	11	g.32413530T>C	c.1435A>G	-	0.0	Het	AD	Missense	VUS/Likely Pathogenic	Unknown ESRD
219	*CUBN*	10	g.16918949T>G	c.9053A>C	rs370778353	0.0001936	Hom	AR	Missense	VUS/Likely Pathogenic	FSGS
223	*PAX2*	10	g.102509528_102509529insG	c.76dup	rs768607170 rs77453353	0.00003590	Het	AD	Frameshift	Pathogenic	FSGS
232	*COL4A4*	2	g.227896862_227896870del	c.3699_3706+1del	-	0.0	Hom	AD/AR	Splice junction loss	Pathogenic	Unknown CKD
235	*ADCK4*	19	g.41206037_41206038insCA	c.1077_1078insTG	-	0.0	Hom	AR	Frameshift	Pathogenic	Unknown CKD
237	*COL4A4*	2	g.227968749C>T	c.755G>A	-	0.0	Het	AD/AR	Missense	VUS/Likely Pathogenic	FSGS
243	*EMP2*	16	g.10641396C>G	c.78+1G>C	rs747072310	0.00003103	Hom	AR	Splicing Variant	Likely Pathogenic	Unknown CKD
244	*PODXL*	7	g.131241029_131241030insGGGGAC	c.89_90insGTCCCC	-	0.0	Hom	AD/AR	In-frame insertion	VUS/Likely Pathogenic	FSGS

* FSGS: Focal segmental glomerulosclerosis, CKD: Chronic kidney disease, ESRD: End Stage Renal Disease, TX: Transplant, AR: Autosomal Recessive, AD: Autosomal Dominant.

**Table 3 ijms-25-01436-t003:** Variant of Unknown Significance in Nephrotic Syndrome-related genes found in our cohort.

Sample	Candidate Gene Name	Chromosome	Genomic Variation	cDNA Variation	dbSNP	Allele Frequency	Inheritance	Genotype	Function	ACMG	Phenotype *
48	PLCE1	10	g.95987105A>G	c.1852A>G	rs200180170	0.0004975	AR	Het	Missense	VUS	Unknown ESRD
60	MYO1E	15	g.59450575G>A	c.2789C>T	-	0.0	AR	Het	Missense	VUS	Unknown ESRD
92	COQ6	14	g.74428075T>C	c.1091T>C	rs747211443	0.00001548	AR	Het	Missense	VUS	Unknown ESRD
170	PLCE1	10	g.96013948G>C	c.3281G>C	rs61732523	0.0002567	AR	Het	Missense	Pathogenic (VUS meaning)	Unknown ESRD
			g.95931087G>T	c.1643G>T	rs17417407	0.1667	AR	Het	Missense	Benign (VUS meaning)	
191	NPHS1	19	g.36340506A>C	c.658T>G	rs115333628	0.001827	AR	Het	Missense	VUS/conflicting interpretation of pathogenicity based on ClinVar	Unknown ESRD
			g.36342212C>T	c.349G>A	rs3814995	0.3102	AR	Het	Missense	Benign	

* ESRD: End Stage Renal Disease; AR: Autosomal Recessive; Het: Heterozygous; VUS meaning: it has been reported to manifest the final explanation of manual curation during the prioritization process, and is not only expressed in online tools of pathogenicity, such as ACMG and others.

**Table 4 ijms-25-01436-t004:** PAX2 pathogenic variants in a family with kidney diseases.

Sample	Gene	Chrom	Genomic Variation	cDNA	dbSNP	Allele Frequency	Genotype	Inheritance	Functional Effect	ACMG Interpretation	Phenotype
223	PAX2	10	g.102509528_102509529insG	c.76dup	rs768607170	0.00003590	Het	AD	Frameshift	Pathogenic	FSGS
224	PAX2	10	g.102509528_102509529insG	c.76dup	rs768607170	0.00003590	Het	AD	Frameshift	Pathogenic	MPGN II
225	PAX2	10	g.102509528_102509529insG	c.76dup	rs768607170	0.00003590	Het	AD	Frameshift	Pathogenic	Unknown ESRD

**Table 5 ijms-25-01436-t005:** GLA gene mutations identified in the analysed cohort.

ID Sample	GLA Genomic Variation (Chromosome X)	Genotype	GLA cDNA Variation	ACMG Interpretation	Notes	Phenotype *
5	g.100653420C>A	HET	c.937G>T; p.Asp313Tyr	Pathogenic	exon 6 of 7 position 136 of 198 (coding)	Unknown CKD
7	g.100662901G>A	HET	c.-10C>T	Benign	exon 1 of 7 (5′UTR) position 101 of 304	FSGS
11	g.100662901G>A	HET	c.-10C>T	Benign	exon 1 of 7 (5′UTR) position 101 of 304	Unknown CKD
38	g.100662901G>A	HOM	c.-10C>T	Benign	exon 1 of 7 (5′UTR) position 101 of 304	FSGS
41	g.100662901G>A	HET	c.-10C>T	Benign	exon 1 of 7 (5′UTR) position 101 of 304	Unknown ESRD (TX)
43	g.100662901G>A	HOM	c.-10C>T	Benign	exon 1 of 7 (5′UTR) position 101 of 304	Unknown ESRD (TX)
46	g.100662901G>A	HET	c.-10C>T	Benign	exon 1 of 7 (5′UTR) position 101 of 304	Unknown ESRD (TX)
65	g.100662903C>T	HET	c.-12G>A	Benign	exon 1 of 7 (5′UTR) position 11 of 216	FSGS
93	g.100662903C>T	HET	c.-12G>A	Benign	exon 1 of 7 (5′UTR) position 11 of 216	FSGS
133	g.100662901G>A	HOM	c.-10C>T	Benign	exon 1 of 7 (5′UTR) position 101 of 304	Unknown ESRD (TX)
148	g.100662901G>A	HOM	c.-10C>T	Benign	exon 1 of 7 (5′UTR) position 101 of 304	Unknown ESRD (TX)
152	g.100662901G>A	HOM	c.-10C>T	Benign	exon 1 of 7 (5′UTR) position 101 of 304	Unknown ESRD (TX)
153	g.100662903C>T	HOM	c.-12G>A	Benign	exon 1 of 7 (5′UTR) position 11 of 216	Unknown ESRD (TX)
167	g.100652828G>T	HET	c.1259C>A; p.Thr420Lys	VUS/Likely Pathogenic	exon 7 of 7 position 260 of 297 (coding)	Unknown ESRD (TX)
169	g.100662901G>A	HET	c.-10C>T	Benign	exon 1 of 7 (5′UTR) position 101 of 304	Unknown ESRD (TX)
176	g.100662901G>A	HET	c.-10C>T	Benign	exon 1 of 7 (5′UTR) position 101 of 304	Unknown ESRD (TX)
190	g.100662901G>A	HET	c.-10C>T	Benign	exon 1 of 7 (5′UTR) position 101 of 304	FSGS
192	g.100662901G>A	HOM	c.-10C>T	Benign	exon 1 of 7 (5′UTR) position 101 of 304	Unknown CKD
211	g.100653420C>A	HET	c.937G>T; p.Asp313Tyr	Pathogenic	exon 6 of 7 position 136 of 198 (coding)	Unknown ESRD (TX)
	g.100662901G>A	HET	c.-10C>T	Benign	exon 1 of 7 (5′UTR) position 101 of 304	Unknown ESRD (TX)
217	g.100662903C>T	HOM	c.-12G>A	Benign	exon 1 of 7 (5′UTR) position 11 of 216	Unknown ESRD (TX)
223	g.100662903C>T	HOM	c.-12G>A	Benign	exon 1 of 7 (5′UTR) position 11 of 216	FSGS
224	g.100662903C>T	HET	c.-12G>A	Benign	exon 1 of 7 (5′UTR) position 11 of 216	Unknown ESRD (TX)
226	g.100662903C>T	HET	c.-12G>A	Benign	exon 1 of 7 (5′UTR) position 11 of 216	Unknown CKD

* FSGS: Focal segmental glomerulosclerosis, CKD: Chronic kidney disease, ESRD: End Stage Renal Disease, TX: Transplant.

## Data Availability

Data is contained within the article and Appendix A.

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
