# Peer review of "What Is Hidden in Patients with Unknown Nephropathy? Genetic Screening Could Be the Missing Link in Kidney Transplantation Diagnosis and Management"

_ijms, 2024, doi:10.3390/ijms25031436_

Round 1

Reviewer 1 Report

Comments and Suggestions for Authors

This research is a single center observational, prospective study. Selected patients were affected by CKD or ESRD, but without histological diagnosis of the cause of nephropathy. Also, patients with biopsy confirmed idiopathic FSGS, were included in the study. The authors wanted to investigate whether genetic screening by Next Generation Sequencing (NGS) could help identify the primary renal defect associated with CKD/ESRD and FSGS.

This is interesting and useful research.

My comments mainly relate to the Result section. The presentation of the results have to be improved. Tables and Figures were cited in the text, but these were not icluded in the manucript. Also, the order of Tables and Figures should be checked. Figure 1 and Table 1 were cited at the end of the manuscript.

It would be appropriate to include some general information about the NGS approach.

In Introduction section, NGS abbrevation, should be written in full name when is used for the first time in the text (line 63).

Comments on the Quality of English Language

Minor editing of English language required.

Author Response

Specific comments to Reviewer 1:

This research is a single center observational, prospective study. Selected patients were affected by CKD or ESRD, but without histological diagnosis of the cause of nephropathy. Also, patients with biopsy confirmed idiopathic FSGS, were included in the study. The authors wanted to investigate whether genetic screening by Next Generation Sequencing (NGS) could help identify the primary renal defect associated with CKD/ESRD and FSGS.

This is interesting and useful research.

We thank the reviewer for the effort and time spent to revise the manuscript.

My comments mainly relate to the Result section. The presentation of the results have to be improved. Tables and Figures were cited in the text, but these were not icluded in the manucript. Also, the order of Tables and Figures should be checked. Figure 1 and Table 1 were cited at the end of the manuscript.

As suggested by this reviewer, we revised the description of results. In particular, the workflow used to select patients and the description of patient’s characteristics is now included in results in order to better clarify patient population. Table 1 and figure 1 have been moved in the first paragraph of results.

It would be appropriate to include some general information about the NGS approach.

In Introduction section, NGS abbrevation, should be written in full name when is used for the first time in the text (line 63).

In the novel version of the manuscript we included a description of the use of NGS for disease diagnosis, prognosis, therapeutic decision-making, and patient follow-up (line63). We also modified NGS abbreviation with full name.

Reviewer 2 Report

Comments and Suggestions for Authors

Limitations of the study should be listed in the Discussion.

It is a very good paper and my only comment is that limitations of the study should be added by the Authors.

1. The main question is whether NGS may be helpful in establishing of the diagnosis leading to CKD, and, subsequently, to ESKD. The authors showed that analysis with NGS allows to establish the diagnosis in approx. 10% of cases with CKD of unknown origin. 2. The results explain the background of CKD in 10% of CKD of unknown origin. Therefore, I find the article original, and I think the results may become clinically useful. 3. Reasons of 10% cases of CKD of unknown origin are clarified. 4. Methodology is correct, no improvements are required. 5. Conclusions arises from the results. Results of NGS explained what were the reasons of CKD in 10% of cases studied. Further studies are required to elucidate the causes of CKD in remaining 90% of cases. 6. References are OK. 7. The quality of data is OK. There are no tables and figures in the manuscript, but I feel they are unnecessary.

Author Response

Specific comments to Reviewer 2:

Limitations of the study should be listed in the Discussion.

It is a very good paper and my only comment is that limitations of the study should be added by the Authors.

We thank the reviewer for the effort and time spent to revise the manuscript. In the novel version of the manuscript we further clarified the limitation of the study in the discussion as reported in lines 325-326, 358-361, 378-380.

  1. The main question is whether NGS may be helpful in establishing of the diagnosis leading to CKD, and, subsequently, to ESKD. The authors showed that analysis with NGS allows to establish the diagnosis in approx. 10% of cases with CKD of unknown origin. 2. The results explain the background of CKD in 10% of CKD of unknown origin. Therefore, I find the article original, and I think the results may become clinically useful. 3. Reasons of 10% cases of CKD of unknown origin are clarified. 4. Methodology is correct, no improvements are required. 5. Conclusions arises from the results. Results of NGS explained what were the reasons of CKD in 10% of cases studied. Further studies are required to elucidate the causes of CKD in remaining 90% of cases. 6. References are OK. 7. The quality of data is OK. There are no tables and figures in the manuscript, but I feel they are unnecessary.

We thank the reviewer for the comments. In the novel version of the manuscript the workflow used to select patients and the description of patient’s characteristics is now included in the first paragraph of results in order to better clarify patient selection. Table 1 and figure 1 have been moved in the first paragraph of results. Other tables and figures are included in the main text.

Round 2

Reviewer 1 Report

Comments and Suggestions for Authors

The authors responded to all comments.

Comments on the Quality of English Language

Minor editing of English language required.